# Reconstruction of 3D Information of Buildings from Single-View Images Based on Shadow Information

**Zhixin Li [1], Song Ji [1,*], Dazhao Fan [1], Zhen Yan [1], Fengyi Wang [2] and Ren Wang [3]**

1. Institute of Geospatial Information, PLA Strategic Support Force Information Engineering University, Zhengzhou 450001, China; lzx1294622314@163.com (Z.L.); fdzcehui@163.com (D.F.); yanzhen7958@163.com (Z.Y.)
2. Coll Surveying & Geoinformat, North China University Water Resources & Elect Power, Zhengzhou 450046, China; 15690860262@163.com
3. Shandong Wuzheng Group Co., Ltd., Rizhao 276800, China; chxy_wangren@163.com
* Correspondence: jisong_chxy@163.com

**Abstract:** Accurate building geometry information is crucial for urban planning in constrained spaces, fueling the growing demand for large-scale, high-precision 3D city modeling. Traditional methods like oblique photogrammetry and LiDAR prove time consuming and expensive for low-cost 3D reconstruction of expansive urban scenes. Addressing this challenge, our study proposes a novel approach to leveraging single-view remote sensing images. By integrating shadow information with deep learning networks, our method measures building height and employs a semantic segmentation technique for single-image high-rise building reconstruction. In addition, we have designed complex shadow measurement algorithms and building contour correction algorithms to improve the accuracy of building models in conjunction with our previous research. We evaluate the method's precision, time efficiency, and applicability across various data sources, scenarios, and scales. The results demonstrate the rapid and accurate acquisition of 3D building data with maintained geometric accuracy (mean error below 5 m). This approach offers an economical and effective solution for large-scale urban modeling, bridging the gap in cost-efficient 3D reconstruction techniques.

**Keywords:** 3D reconstruction; urban 3D probabilistic model; deep learning; building height estimation; shadow length measurement

## 1. Introduction

With the development of modern cities, the tension between the drastic increase in urban populations and a shortage of building plots is growing [1]. With the rapid growth of urban areas, the height of buildings has increased drastically, and high-rise buildings have become the city's landmarks. The investigation and maintenance of high-rise buildings is not only related to the safety and sustainable development of the buildings themselves, but also involves the image of the city, environmental protection, and many aspects of urban development, which is of broad and far-reaching significance and has become the key work of urban planning departments in countries around the world [2].

Three-dimensional city models can facilitate the planning and management of the architectural environment, which leads to a high user demand for this type of data. Despite the growing recognition of the importance of such data, obtaining them in a low-cost and efficient manner remains challenging [3]. With the development of geographic information technology, 3D city models based on satellite remote sensing, oblique photography, light detection and ranging (LiDAR), and other measurement technologies have become increasingly detailed. However, the capability of 3D city modeling based on satellite remote sensing technology is limited, and oblique photogrammetry is relatively time-consuming. The amount of 3D point-cloud data acquired based on LiDAR is staggering, reaching terabytes over large areas, and the acquisition cost is high. By contrast, the 3D reconstruction

of buildings based on single-view images avoids these limitations. It has excellent potential for large-scale non-measurement scene applications and has become an important research direction in recent years [4]. The open sourcing of geospatial data such as satellite images, building footprint vector data, and LIDAR point clouds [5] creates opportunities to generate large-scale 3D city models cost-effectively.

Compared to LiDAR data and multi-view images, the 3D reconstruction of single-view remote sensing images is challenging [4]. The invisibility of building footprints and façade information, severe shadowing effects, and extreme variations in building heights limit the application scenarios for the 3D reconstruction of single-view remote sensing images, with the majority of studies carried out on low-rise buildings. A simple understanding of 3D modeling involves assigning elevation information to 2D data and displaying it in 3D form [6]. The 3D reconstruction of buildings based on single-view images can also assign building height information to 2D building outline data. In recent years, with the rapid development of deep learning, obtaining building outlines has become easier; however, it is still challenging to accurately obtain height information. Although existing building extraction algorithms can obtain building boundaries better, for remote sensing images with non-orthographic projection, the acquired boundaries contain both roofs and façades, which deviate from the real building footprints [7]. In addition, building height increases as the measurement area expands, making it challenging to learn an accurate height value directly through deep neural networks. Shadows in images have many adverse effects on remote sensing interpretation; however, building heights can be calculated based on the length of a building's shadow and its geometric relationship with the sun and the sensor. This method of inferring information regarding the building height by analyzing the shadow cast and its relationship to the ground is called the shadow method of height determination (hereinafter referred to as the "shadow method"), and it provides an effective means of measuring and analyzing the 3D form of a building. The shadow method has some errors in building height estimation because shadows are affected by a variety of factors, such as the occlusion of other features, adhesion between shadows, complex building structures, and terrain undulation, which lead to irregular shapes or challenges in analyzing shadows and affect the accuracy of height estimation [8–11]. Despite some drawbacks and challenges, the shadow method remains a valuable and common method that can be combined with other techniques to improve the accuracy and effectiveness of height estimation.

This paper proposes a new method for the 3D reconstruction of single-view remote sensing images that addresses the above problems, including the poor accuracy and low applicability of shadow-based building height, and the problem that the initial building profile contains roof and façade information. This work also provides new ideas for non-specialized mappers to acquire large-scale urban 3D information, and reduces the need for data, hardware and software, and professional capabilities. It is worth noting that this paper is an extension of our previous work on shadow measurement [12,13], in which we solve the problems of inefficiency and the poor accuracy of shadow length measurement by proposing a method for partitioning shadows, which reduces the data requirements of the algorithm and improves the accuracy and applicability of the algorithm.

The contributions of this paper can be summarized as follows:

1.  A new method for the 3D reconstruction of single-view remote sensing images is proposed. This method combines the advantages of the shadow method and deep-learning semantic segmentation technology to address the challenge of 3D reconstructing high-rise buildings in cities. Compared to the single-view depth estimation method, the method proposed in this study has the advantages of easy reproducibility, reliable precision, low time, and cost-effectiveness.
2.  The complex shadow measurement algorithms and designed building contour correction algorithms are combined to further improve the vertical and level precision of 3D building models.

3. We explore the accuracy, time, and applicability issues of the method for different data sources, scenarios, and scale tasks, as well as the limitations of the shadow method in obtaining building heights.

The remainder of the paper is organized as follows: Section 2 presents studies on estimating building heights from single-view images; Section 3 describes the datasets and methodology used in this study, including the extraction of buildings and shadows, the measurement of building height, and the correction of building outlines; Section 4 reports the qualitative and quantitative results of the method; and Section 5 discusses the validity and applicability of the proposed method. Finally, the conclusions are presented in Section 6.

## 2. Literature Review

Recently, studies on building height estimation using single-view images have attracted significant attention. This task is important for urban planning, remote-sensing analysis, and 3D reconstruction [14]. Owing to the lack of depth information in single-view images, researchers have worked on estimating building heights using methods such as shadow methods and depth estimation.

Since 1989, various aerial photogrammetry researchers have used shadow information to estimate building height [15]. Shettigara et al. [16] used shadow information from SPOT panchromatic images to develop a model and derive building heights. Wang et al. [17] established the geometric relationship between shadow length and building height using ZY3 images by calculating building height in conjunction with the shadow length. Based on this, 3D models of the city buildings were developed. Liasis [9] implemented a new active outline model using spectral and spatial analysis information from satellite images, which optimized the shadow segmentation process of a building, improved the accuracy of shadow extraction, and estimated the building height from the shadow length. Izadi et al. [11] proposed a method for calculating building heights by detecting the boundaries of buildings and shadows and implementing building heights in fast-bird images. Shao et al. [18] proposed a method that combined the object space index of images to improve the precision of shadow extraction and used IKONOS images as an example to estimate building heights with shadow lengths. Xie et al. [19] analyzed scene descriptions by analyzing the shapes of building shadows, distribution density, and regional topography. They classified the building scenes into three major types: ordinary, dense, and complex. Subsequently, a multi-scene building height estimation model was developed to explain the geometric relationship between buildings and shadows in different scenes. Zhao et al. [20] labeled sample building heights using photons collected by ICESat-2 and developed a height estimation model by minimizing the global error for all sample buildings. This complements the shadow-height estimation method. While the shadow method has made significant strides in building height acquisition, shadows in densely built-up areas are prone to be obscured and stick to each other, which makes the accurate measurement of shadow lengths difficult, further leading to unreliable estimates of building heights.

Another approach is to utilize depth-estimation techniques for building height estimation. In recent years, deep learning has also been added to the extraction of feature elevation or height information from aerial or satellite images [21–23], opening up a new avenue for building height estimation. Amirkolaee and Arefi [24] developed a deep CNN to estimate the digital surface model (DSM) from a single aerial image and proved its effectiveness on the ISPRS dataset. Liu et al. [25] proposed an end-to-end trainable convolutional-inverse convolutional deep neural network architecture that allows mapping from a single aerial image to a DSM to analyze city scenes. Ghamisi et al. [26] utilized conditional generative adversarial networks (cGAN) to generate a synthetic DSM from a single aerial image. Bittner et al. [27] used a cGAN to optimize building roof surfaces by mapping DSMs from stereo aerial images to LOD2 detail-level models. Carvalho et al. [28] introduced a multitask learning network to take full advantage of the mutual information of different tasks while dealing with land cover mapping and normalized DSM (nDSM) estimation

and found that multitask learning outperformed single-task learning. This method overcomes the limitations of the shadow method to some extent and improves the accuracy and stability of building height estimation. Obtaining large amounts of accurately labeled single-view image data is time-consuming and expensive. The lack of large-scale standardized datasets and high-quality annotations limits algorithm performance improvement and general-purpose scaling.

In addition to these approaches, other innovative research methods are available. For example, the multitask building reconstruction network developed by Li et al. [4] can retrieve the roof and footprint information of a building from oblique images and estimate the building height based on the offset between them. Yan and Huang [29] combined the vanishing point method with deep learning to design a framework for extracting building height information from a single street-view image, which increased its competitiveness for large-scale building height estimation with minimal input. Sun et al. [30] accomplished large-scale building height retrieval using only a single SAR image and converted the building height retrieval problem into a boundary-box regression problem. The building heights were computed using the positional relationship between the building footprint and its boundary box. More innovative methods and technologies should be applied in this field in the future to further enhance the accuracy and reliability of building height estimation.

## 3. Materials and Methods

### 3.1. Study Area and Data

Three representative study areas in Zhengzhou City, China are selected for this study (Figure 1). Study area (a) has a neat and orderly building plan with moderate density and is mainly dominated by residential buildings. Multistory high-rise buildings with high densities dominate the study area (b). Study area (c) is a commercial area characterized by high density, various types, and a complex distribution of buildings. We use data from China's Gaofen (referred to as "GF") and ZiYuan (referred to as "ZY") series of satellites, such as GF1, GF2, GF7, and ZY3 (Table 1). GF1 has a higher spatial resolution and a shorter revisit period. GF2 reaches a submeter spatial resolution and can capture more detailed features of buildings, thus providing richer data support for building height inversion. In addition, GF7 and ZY3 have high-precision stereo imaging capabilities, which can provide more accurate 3D information about buildings and can be used as reference data for building height assessments. Three-dimensional real-scene models of the study area (c) are collected simultaneously. The applicability and performance of the method can be fully assessed by selecting study areas and data with different characteristics. The data used in this paper are downloaded from the Natural Resources Satellite Remote Sensing Cloud Service Platform (http://114.116.226.59/chinese/normal/ (accessed on 23 September 2023)).

**Table 1.** Data details.

| Data | Area | Resolution/m | Time |
| --- | --- | --- | --- |
| GF1 | b | 2 | 25 June 2020 11:25:38 |
| GF1 | b, c | 2 | 11 September 2020 11:24:28 |
| GF1 | b | 2 | 30 October 2021 10:57:59 |
| GF1 | b | 2 | 12 January 2021 11:18:18 |
| GF1 | b | 2 | 8 April 2021 11:11:03 |
| GF1 | b | 2 | 14 March 2023 10:50:49 |
| GF2 | c | 0.8 | 7 June 2020 11:28:33 |
| GF7 | a, b, c | 0.65 | 20 September 2020 11:31:19 |
| ZY3 | c | 2.1 | 4 September 2020 11:17:58 |
| 3D model | c | 0.1 | 12 November 2021 |

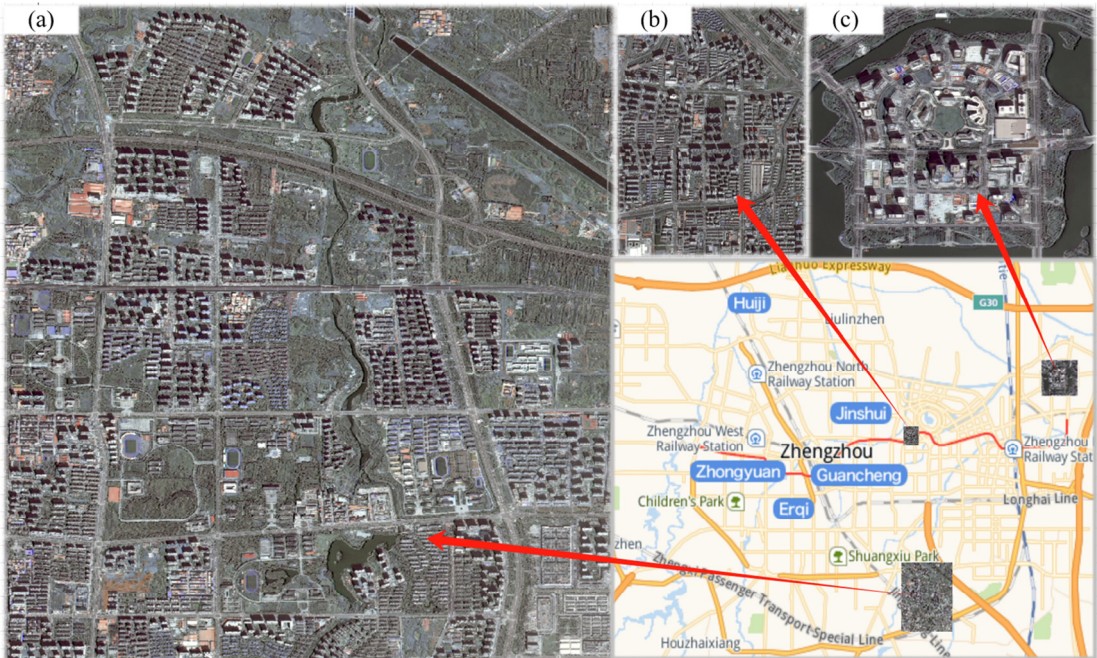

**Figure 1.** Research area. (**a**) Study area a; (**b**) Study area b; (**c**) Study area c.

### 3.2. Methodology

The 3D information of a city is typically constructed using vectorized architectural data coupled with urban surface topography data, which are much less expensive to store and compute than 3D real-scene models and laser point clouds. An abstract generalization of a real cityscape is performed to eliminate or suppress unimportant details and enhance and highlight important information in the data. The most direct manifestation of this is that the model expresses only two major elements: urban surface and buildings, eliminating elements such as vegetation, water bodies, and aboveground facilities that have less impact and are not amenable to data collection. This data model, which consists of data representing the urban terrain and vector building data with building heights, is referred to as an urban 3D probabilistic model.

The urban 3D probabilistic model has relative height, which refers only to the building height and excludes topographic elevation, and absolute height, which reflects the actual height of the city model. Urban terrain elevation can be provided by an open-source DEM [31]. Thus, obtaining the footprints and heights of buildings from single-view satellite remote sensing images is the focus of this study. As shown in Figure 2, the steps of the method used in this study are as follows:

To extract building shadows, remote sensing images are pretreated with radiation correction, geometric correction, and image fusion to eliminate color differences and image distortions caused by atmospheric scattering, sensor attitude, and earth curvature. Semantic segmentation models are used to extract buildings and their shadows and utilize the topological relationship between the two to eliminate erroneous results.

The heights of the buildings are measured using the shadow method. The traditional building height measurement model is improved by removing complex and less influential parameters to enable more automated building height measurements. In addition, the use of building outline data to improve the measurement of shadow length not only improves the accuracy of shadow measurement but also addresses the measurement problem in the case of incomplete shadows and allows for more accurate information regarding the height of the building.

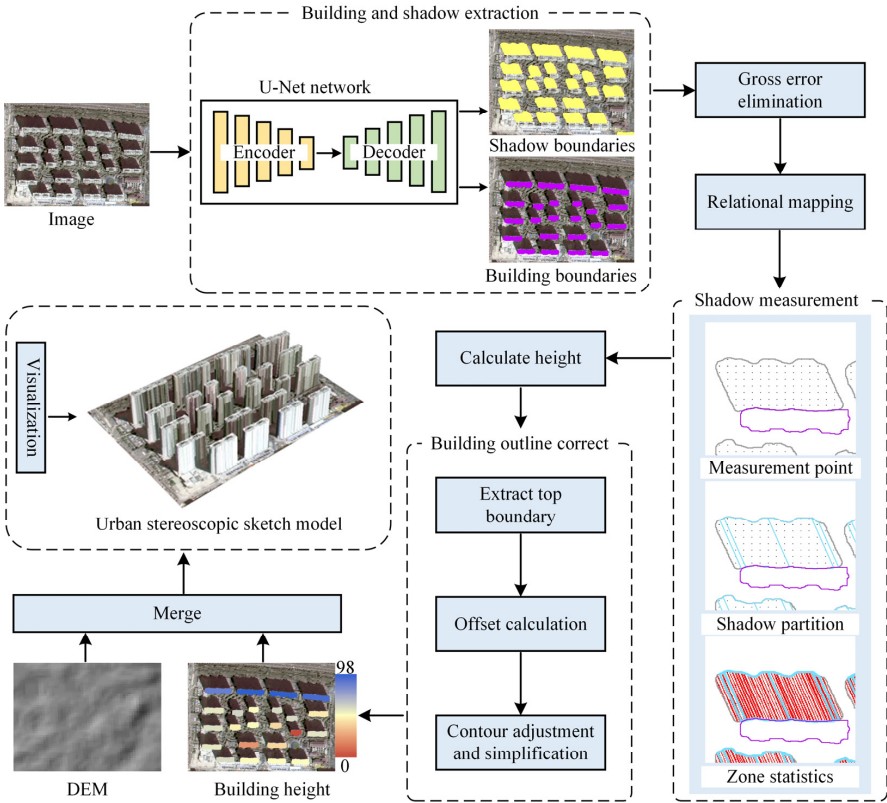

**Figure 2.** Flowchart of proposed method.

The initial building outline contains the walls, top and side edges of the roof, and the bottom outline of the building. A building top-edge outline offset correction model is constructed, and a building outline correction algorithm is designed to convert the building outline to the actual building footprint.

Combined with the theory of high-precision single-image positioning, the accuracy, performance, and applicability of the proposed method for different data sources, scenarios, and task scales are analyzed and validated.

Sections 3.4 and 3.5 describe details of building height calculations and footprint acquisition.

### 3.3. Extraction of Buildings and Shadows

Semantic segmentation models, such as UNet, SegNet, and DeepLab are typically employed to extract buildings and shadows. In this study, UNet is selected as the detection model. This is because UNet is a simple and effective convolutional neural network architecture that has been widely validated and applied to building and shadow extraction tasks in various scenarios [32]. Its encoder–decoder architecture and cross-layer connection mechanisms allow UNet to capture the details and contextual information of buildings and their shadows accurately [33].

A total of 1950 images (512 × 512 pixels) are labeled from high-resolution satellite images captured from different sensors to train the U-Net model for boundary extraction. Figure 3a,b shows the images corresponding to the two samples. Figure 3c,d displays the labeled buildings and shadow boundaries, respectively. A data augmentation method is used to enlarge the training samples before network training. The images are then randomly divided into training, testing, and validation sets at a ratio of 7:2:1. The U-Net model for boundary extraction is constructed for 150 rounds on the training set with an initial learning rate of 0.001 and a batch size of 16. Buildings and shadows exhibit significant differences in feature representation. The two types of targets are trained separately to better model their respective features and attributes and avoid model performance degradation owing

to excessive or unbalanced differences. To address challenges arising from similar spectral characteristics among water bodies, non-building shadows, and building shadows, and the limitation of the shadow method in obtaining height information for buildings without or with minimal shadows, a necessity emerges to implement a filtering process based on the topological relationship between buildings and shadows.

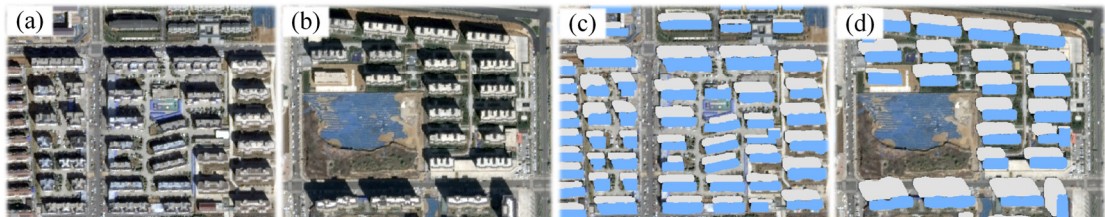

**Figure 3.** Sample data: (**a**,**b**) display the training images; (**c**,**d**) display the buildings and shadow labels.

### 3.4. Measurement of Building Height

3.4.1. Calculation of Shadow Length

Shadow length calculations directly impact the estimation of building heights. Shadow length measurements in remote-sensing images face many difficulties and challenges and are complicated by changes in illumination conditions, the complexity of terrain and features, image resolution limitations, shadow lengths, and pattern changes in different periods, seasons, or weather conditions. In addition, shielding and obfuscation are puzzling; other features or buildings may partially or entirely block shadows, making it difficult to measure shadow lengths accurately. To address these challenges, we utilize the complex shadow measurement algorithm proposed in a previous study to solve the problem [12]. This algorithm is designed to accommodate the measurement requirements for various types of shadows. The shadow measurement process is shown in Algorithm 1.

3.4.2. Measurement of Building Height

Shadow-based building height estimation models are usually categorized into two types based on the geometric relationship between the sun and the sensor. Figure 4a illustrates the case where the sun is located on the same side as the sensor, and part of the shadow is blocked by the building. Figure 4b illustrates the case where the sun is located on the opposite side of the sensor, and the sensor captures the complete shadow of the building. The building height $H$ is related only to the shadow length and solar elevation angle, as shown in Equation (1). The building height is related to the solar azimuth, satellite elevation angle, and satellite azimuth, in addition to the shadow length and solar elevation angle [13,19]. See Equation (2).

---

**Algorithm 1**: Building shadow measurement

---

**Input:** $B$ = Building contours, $S$ = Shadow mask
**Output:** $L$ = Shadow length
***Lines*** ← Distribution point measurement ($S$)
For $i$ in $B$:
    *Areas*← Shadow partition ($i,S$)
    $Ls = [\cdots]$
    For *Area* in *Areas*:
        *Line_A* ← Partition statistics (*Area*, *Lines*)
        *Line_A* ← Gross error elimination (*Line_A*)
        $L\_A$ ← Compute optimal value (*Line_A*)
        $Ls$ append $L\_A$
    Endfor
    $L$ ← Optimal value evaluation($Ls$)
Endfor

---

$$H = L \cdot \tan \beta = L \cdot k_1 \tag{1}$$

$$H = \frac{L_2 \cdot \tan \beta \cdot \tan \alpha \cdot \sin(\xi - \gamma)}{\tan \alpha \cdot \sin(\xi - \gamma) - \tan \beta \cdot \sin(\xi - \varepsilon)} = L_2 \cdot k_2 \tag{2}$$

where $\xi$ denotes the azimuth angle of the building and $k_1$ and $k_2$ are scale factors.

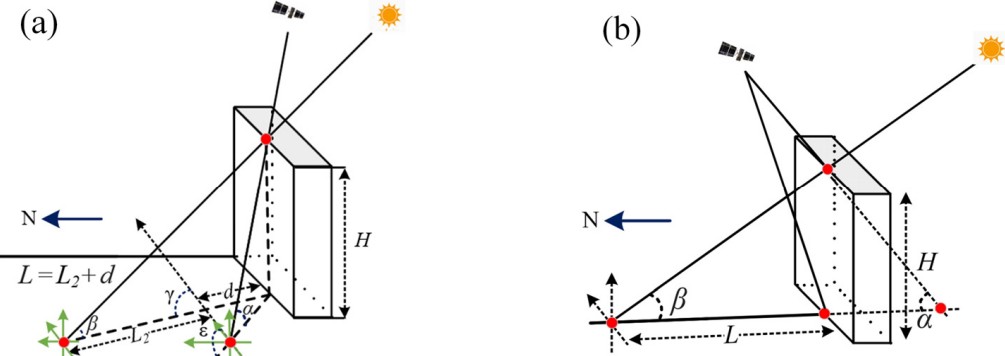

**Figure 4.** Schematic view of the geometric relationship between buildings and shadows. (**a**) Sensor on same side as sun. (**b**) Sensor on opposite side of sun.

In Figure 4, $\beta$ is the solar elevation angle, $\gamma$ is the solar azimuth angle, $\alpha$ is the sensor elevation angle, and $\varepsilon$ is the sensor azimuth angle. $L$ is the actual length of the building shadow, $L_2$ is the length of the building shadow observed in remote sensing images, and $d$ is the length of the building shadow occluded by the building body.

The purpose of this study is to fulfill the needs of the task by using only the information provided by a single image. In this case, the elevation and azimuth angles of the sun and the satellite can be obtained from the image metadata. Building azimuth calculations are complex and have little effect on the estimated building height results. Therefore, adjust Equation (2) to remove the building orientation parameter, and the result is shown in Equation (3). At the same time, the reference direction for all azimuths is set to the east for ease of calculation and programming.

$$H = \frac{L_2 \cdot \tan \beta \cdot \tan \alpha \cdot \sin \gamma}{\tan \alpha \cdot \sin \gamma - \tan \beta \cdot \sin \varepsilon} \tag{3}$$

*3.5. Adjustment of Building Outline*

3.5.1. Model of Building Outline Offsets

Typically, remote sensing building images are not orthorectified. Owing to the building height and influence of the sensor, there is a certain amount of offset between the roof and footing of the building when the image is captured, as shown in Figure 5, where the black line segments are the offsets of the building. To correct the building roof offset more conveniently and quickly, this study establishes a simple offset calculation model based on the relationship between the building height and satellite altitude angle while considering only the building roof and bottom offsets.

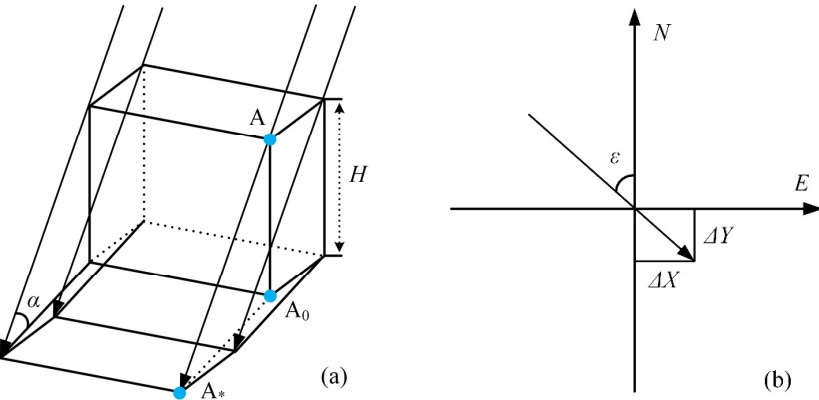

**Figure 5.** (**a**) Building imaging model. (**b**) View of building vertex offset.

In Figure 5, A is the actual building roof point, $A_0$ is the actual building bottom point, and A* is the building roof point at the time of imaging. For the building roof point A, a certain amount of offset occurs during imaging because of the influence of the building height and elevation angle; the actual imaging position of vertex A is not on $A_0$ but on A*. This offset is a constant value independent of the solar attitude for parallel projection. This is only related to the building height and satellite attitude at the instant of the shot, as shown in Equation (4). The components of the building offset in the E and N directions are shown in Figure 5b.

$$\begin{cases} \Delta X = H \cdot sin\varepsilon / tan\alpha \\ \Delta Y = H \cdot cos\varepsilon / tan\alpha \end{cases} \tag{4}$$

The offset of a building can be obtained from its height $H$, satellite elevation angle, and azimuth angle, and the precision of the measurement of the building height determines the precision of the offset solution.

### 3.5.2. Correction of Building Outline

The initial outline obtained from semantic segmentation contains the building's roof, side, and bottom information. In this study, based on the above model of building-outline offsets, a correction algorithm is designed to adjust the building outline to the real building footprint based on the offsets obtained from the building height, as shown in Figure 6. The steps are as follows:

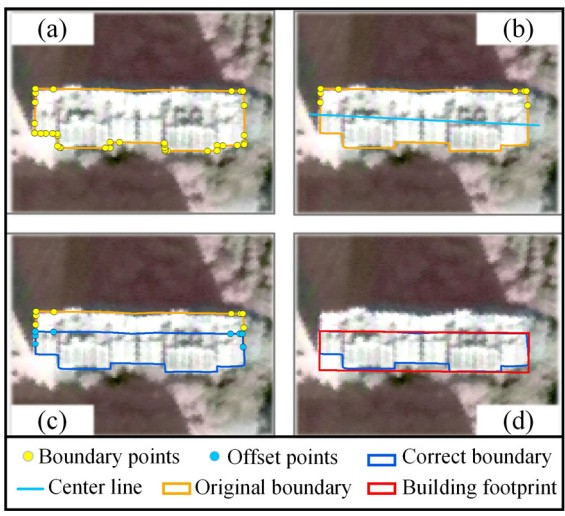

- ● Boundary points ● Offset points ▭ Correct boundary
- ── Center line ▭ Original boundary ▭ Building footprint

**Figure 6.** Process of building outline correction. (**a**) Extraction of boundary points. (**b**) Calculation of building principal directions. (**c**) Classification and correction of boundary points. (**d**) Simplification of outline.

The boundary points are extracted. The building outline is represented as a closed curve of discrete points, denoted *C*. To reduce the computational cost, the vector method is utilized to filter the relatively smooth points in *C*. The points in *C* are assumed to be ordered, and the *i*th point is denoted by $P_i$, where *i* = 1, 2, ..., n. The vector of each edge is calculated using Equations (5) and (6). The angle of each boundary point is calculated using Equation (7), and an angle threshold $\theta$ is set. All the points that satisfied $\theta_i > \theta$ are retained, and these are considered the endpoints of the building outline, denoted as the set *E*. The other points are considered relatively smooth and are discarded.

$$v1_i = P_i - P_{i-1} \tag{5}$$

$$v2_i = P_i - P_{i+1} \tag{6}$$

$$\cos \theta_i = (v1_i \cdot v2_i) / (|v1_i| \cdot |v2_i|) \tag{7}$$

The building main direction *B_v* is calculated using principal component analysis. *E* is constructed as a 2 × m matrix M

$$M = \begin{bmatrix} x_1 & x_2 & \dots & x_m \\ y_1 & y_2 & \dots & y_m \end{bmatrix} \tag{8}$$

The mean $\overline{x}$, $\overline{y}$ of the set *E* are calculated. The $\overline{x}$ and $\overline{y}$ are subtracted from the matrix M, and the homogenized matrix M is yielded.

$$M = \begin{bmatrix} x_1 - \overline{x} & x_2 - \overline{x} & \dots & x_m - \overline{x} \\ y_1 - \overline{y} & y_2 - \overline{y} & \dots & y_m - \overline{y} \end{bmatrix} \tag{9}$$

The covariance *C* is calculated using the matrix M:

$$C = \frac{1}{m} M \cdot M^T \tag{10}$$

All eigenvalues and eigenvectors of the covariance matrix *C* are calculated. Because M is a two-dimensional square matrix, two eigenvalues and their corresponding eigenvectors *c*1 and *c*2 are obtained. Thus, *B_v* is max (*c*1, *c*2).

Classifying and correcting boundary points: The points in set *E* are classified as the roof and bottom points based on the cross-product method. *A* straight line is determined using the geometric center of the building as the origin and *B_v* as the slope. *A* and *B* are assumed to be two points on the line, and *P* is the point to be judged and substituted into Equation (11). The outline endpoints are classified by judging the relative position of point *P* on line *AB* according to the properties of the cross product, both positive and negative.

$$S(A, B, P) = (Ax - Bx) \cdot (Py - By) - (Ay - By) \cdot (Px - Bx) \tag{11}$$

where $S(A, B, P)$ is the cross-product result, points with *S* greater than zero are categorized as roof points, and points with S less than zero are categorized as bottom points.

The offsets obtained using Equation (4) correct the roof points, forming a new building outline with bottom ends. At the intersection of the fixed outlines, the endpoints are sharp and are further filtered using the angular information.

We simplify the building outline and consider its minimum bounding rectangle to be the building footprint. Large-scale city studies require appropriate simplification of the models. Utilizing the minimum bounding rectangle can simplify shape representation, reduce complexity, and preserve the basic boundary information of the building. This simplified building representation is useful for urban planning, GIS analysis, and other large-scale city data processing applications.

*3.6. 3D Reconstruction of Buildings*

An urban 3D probabilistic model is obtained by combining vector data with building height and terrain data. We overlay a digital orthophoto map (DOM) as texture on the model and visualize it with the help of geographic information platforms such as ArcGIS and QGIS. The cost of producing an urban 3D probabilistic model is significantly lower than that of producing an oblique photogrammetric model. Although oblique photogrammetry is becoming less expensive, achieving full-area coverage is challenging. The model is well-suited as a flat replacement for the oblique photogrammetric model to fill in areas not covered by oblique photogrammetry and requires less precision.

*3.7. Evaluation Index*

In this study, the proposed method is evaluated in terms of both the horizontal positioning and vertical precision of a building. Horizontal precision is studied in conjunction with the theory of high-precision single-image positioning. High-precision single-image positioning utilizes a single image to achieve precise localization. First, the SIFT feature-matching technique is used to match the image to be measured with the reference image. Next, we calculate the geometric centers of mass between the building footprints extracted using this method and the reference footprint. Finally, the accuracy of the horizontal localization is evaluated by calculating the deviation between the centers of mass, as shown in Equation (12). The absolute error, mean absolute error (MAE), and root-mean-square error (RMSE) are selected as evaluation indices for building height accuracy. The specific calculation formula is as follows:

$$\Delta_b = \sqrt{(x - x_0)^2 + (y - y_0)^2} \tag{12}$$

$$\text{MAE} = \frac{1}{m}\sum_{i=1}^{m}|H_i - H_{0i}| \tag{13}$$

$$\text{RMSE} = \sqrt{\frac{1}{m}\sum_{i=1}^{m}(H_i - H_{0i})^2} \tag{14}$$

where $\Delta_b$ is the geometric center-of-mass deviation of the building, $x$ and $y$ are the horizontal coordinates of the building footprint, and $x_0$ and $y_0$ are the horizontal coordinates of the reference building footprint. $H_i$ is the value of the building height proposed by the method and $H_{0i}$ is the value of the reference building height.

## 4. Experiments and Analysis
*4.1. Building and Results of Shadow Extraction*

GF7 is a civil stereo mapping satellite with the highest spatial resolution in China, and this study first researches the 3D models of buildings produced by GF7 single-view images. Figure 7 shows the building and shadow measurements of the study area (a). Figure 7a shows the building boundaries extracted from the building extraction model, and Figure 7b shows the shadow boundaries extracted from the shadow extraction model, with areas marked by red boxes to further demonstrate the extraction. Figure 7c shows the GF7 image, and Figure 7d,e shows the extracted building and shadow boundaries. Both buildings and shadows are well extracted, and water bodies and non-building shadows are well suppressed, which can be further used for building height measurements.

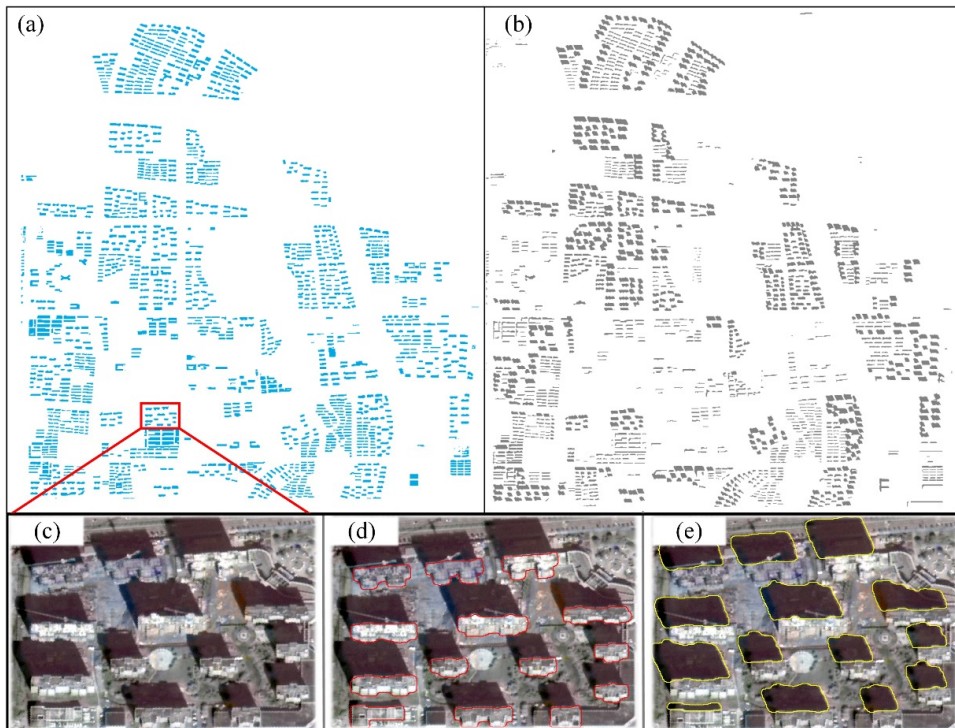

**Figure 7.** (**a**) Building boundary extraction results. (**b**) Shadow extraction; and (**c**) GF7 image of the magnified region (red box). (**d**) Building boundaries extraction results corresponding to (**c**). (**e**) Shadow extraction results corresponding to (**c**).

To quantitatively assess the building and shadow boundaries, metrics such as overall accuracy (OA), precision (P), intersection of union (IOU), F1 value (F), and recall (R) are used [20]. Table 2 presents the statistical results for the test set. In general, the overall precisions of the buildings and shadows extracted using this method are 92.37% and 96.87%, respectively. Moreover, the OA, P, IOU, F, and R values of shadows are higher than those of buildings because the spectral features of shadows are simpler and more accessible for training than those of buildings. It is worth noting that the building boundary extraction model can have such high performance, on the one hand, because it only labels buildings with shadows. By discarding buildings without shadows, such as factories and bungalows, the training difficulty of the model is reduced, and the detection precision of the model is improved. Meanwhile, on the other hand, overly strict labeling may lead to the overfitting of the model, which is a potential reason for the outstanding model performance. The extraction of buildings and shadows is the basis for building height measurements and is not the focus of this study. Therefore, the details of the model are not thoroughly explored.

**Table 2.** Accuracies of building boundaries and shadow extraction.

| Task | OA (%) | P (%) | IOU (%) | F (%) | R (%) |
|---|---|---|---|---|---|
| Building boundaries | 92.37 | 90.59 | 80.77 | 86.36 | 83.14 |
| Building shadow | 96.87 | 95.81 | 90.32 | 94.01 | 92.26 |

*4.2. Qualitative Analysis of Urban 3D Probabilistic Model*

To enhance the precision of the shadow length measurement, this study uses partitioned measurement, and Figure 8 demonstrates the measurement process of this method in the study area (a). It is clear from the figure that each shadow is divided into four partitions, and each section corresponds to one measurement value. For occluded building shadows, the ratio of the measured values of the complete shadow length to the entire set of measured values is low. The percentage of correct measurements can be increased

significantly by partitioning the measurements, thereby effectively reducing the algorithm's requirement for shadow completeness.

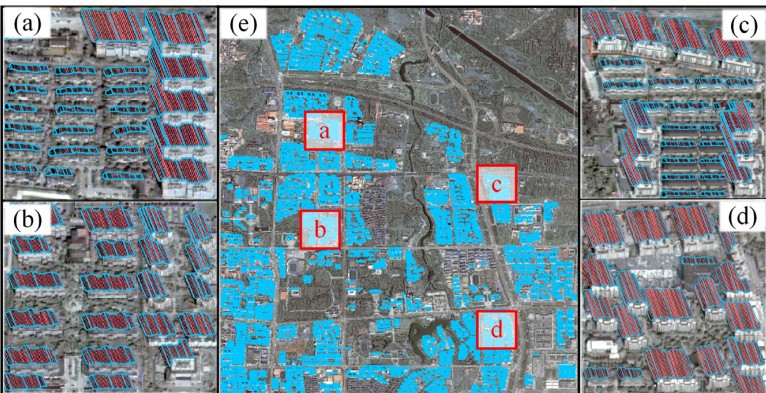

**Figure 8.** Results of shadow length measurement visualization. (**a**–**d**) Local detail maps corresponding to the red rectangular boxes. (**e**) Shadow length measurements in study area a.

Figure 9g shows the results of this method for the 3D reconstruction of buildings in the study area (a), which is visualized with the help of Arcscene. Figure 9a–c shows the reconstruction results of the proposed method for localized details, and Figure 9d–f shows the reconstruction results of multi-view dense matching on the same localized information. By comparing the visualization results of dense matching, it can be observed that the model reconstructed using this method has the advantages of better visualization and no obvious coarseness, which is more in line with the actual situation, and the integrity of the building reconstruction is higher.

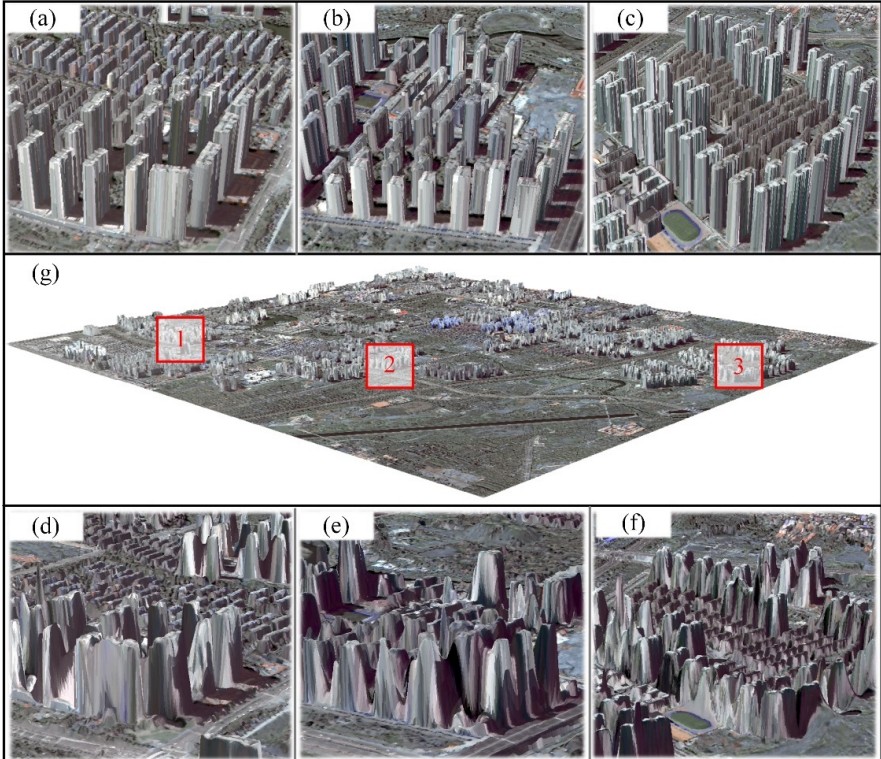

**Figure 9.** Results of the 3D visualization of the building. (**a**,**d**) Local detail maps corresponding to the red rectangular box of No. 1. (**b**,**e**) Local detail maps corresponding to the red rectangular box of No. 1. (**c**,**f**) Local detail maps corresponding to the No. 1 red rectangular box. (**g**) 3D reconstruction results of buildings in study area a.

*4.3. Quantitative Analysis of Urban 3D Probabilistic Models*

4.3.1. Accuracy of Building Height

In this study, a digital surface model (DSM) of the study area (a) using MASI, a commercial 3D reconstruction software, is combined with GF7 and ZY3 data. A normalized DSM (nDSM) is generated using the method proposed by Wu et al. (2022) [34], and the maximum value of nDSM pixels within the building boundary is counted as the building reference height. Each is checked to ensure the accuracy of the reference building heights. Figure 10 illustrates the precision of the building height estimation for the study area (a). Figure 10a shows the frequency histogram of the residuals of the building height estimation, which is the result of excluding statistics other than 3σ from the original residuals. As shown in Figure 10a, the residuals satisfy a normal distribution, with most (75%) of the residuals falling between −5 and 5 m, and there is a slight overall overestimation. From the residual line graph in Figure 10b, it can be seen that there is an overestimation of building heights below 30 m, an underestimation above 70 m, and good performance between 30 and 70 m. This phenomenon occurs because the shadows of the lower buildings are blended with the shadows of vegetation, resulting in longer shadow lengths, whereas the shadows of the higher buildings are overlaid on other features, resulting in shorter shadow lengths. Figure 10c shows the residual distribution of the building heights.

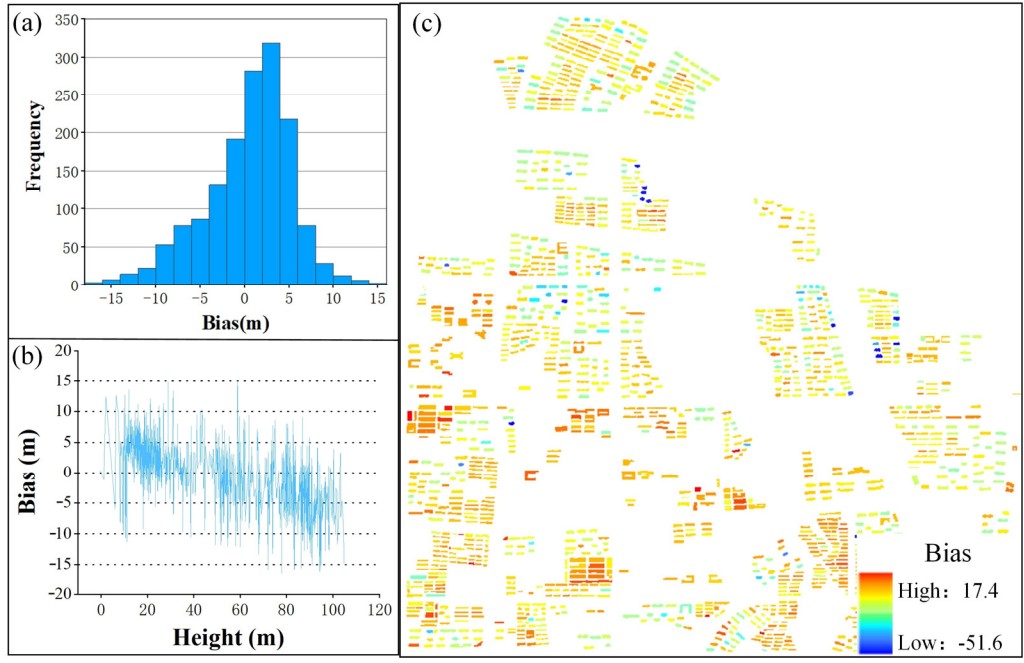

**Figure 10.** (**a**) Frequency histogram of the building height estimation residuals. (**b**) Line chart of the building height estimation residuals. (**c**) Residual distribution of the buildings.

Shadow-based algorithms are implemented to evaluate the performance of the method in this study [9,19]. The main difference between existing shadow-based algorithms is the counting of the shadow measurements. Liasis and Stavrou [9] used the median of the set of shadow measurements as the shadow length, while Xie et al. [19] excluded measurements other than 3σ and took the average of the remaining measurements as the shadow length of the building. Figure 11 illustrates the scatter plot between the estimated and reference building heights in the study area (a) for the different methods. It can be seen that the method in this paper has a better fitting effect, with the highest $R^2$ and relatively low MAE and RMSE for the estimated building heights, with an MAE of 3.852 m and an RMSE of 4.825 m. This benefits from the shadow partitioning measurement algorithm used in this study, which reduces the algorithm's requirement for shadow integrity and improves the accuracy of the shadow length measurement.

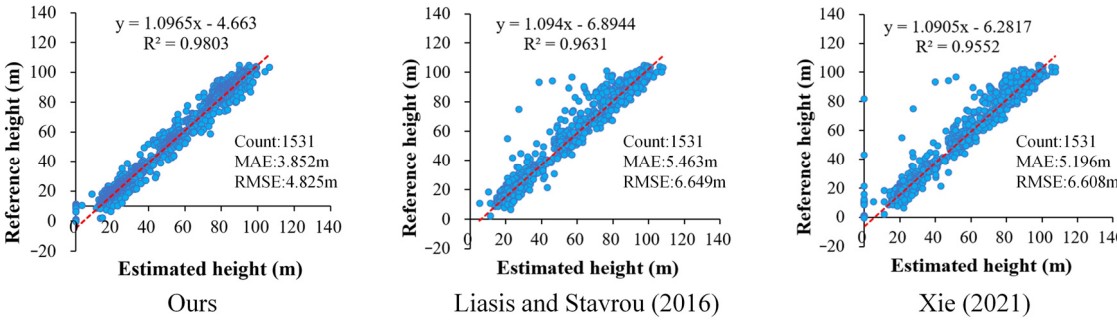

**Figure 11.** Statistical results of error by different methods [9,19].

#### 4.3.2. Precision of Building Horizontal Positioning

Building outlines obtained from high-resolution satellite images differ from those obtained from building footprints. The GF1, GF2, and ZY3 down-view images are essentially vertical, with small offsets of building vertices. The GF7 rearview image has a dip angle of approximately 5°, which leads to a severe building offset in the image. Therefore, this study focuses on correcting the building outline obtained from GF7. The real building footprints are not visible in the remote sensing images; thus, the building roof boundaries in the study area (b) are labeled as a reference. The building outline is changed to a roof outline, and the geometric center-of-mass deviation is counted from the reference boundary, as shown in Figure 12. For study area (c), the real building footprints are outlined directly from the digital orthophoto map produced by the 3D real-scene model. Then, registering GF7 of study (c) to the digital orthophoto map removes the systematic errors between the images. Figure 13 illustrates the building outline correction results for the study area (c).

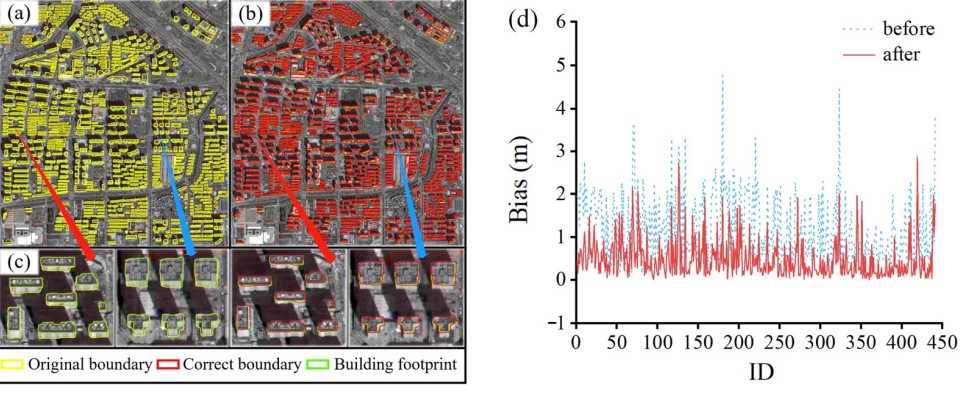

**Figure 12.** (**a**) Study area b before adjustment. (**b**) Study area b after adjustment. (**c**) Local detail diagram before and after adjustment. (**d**) Map of horizontal positioning accuracy.

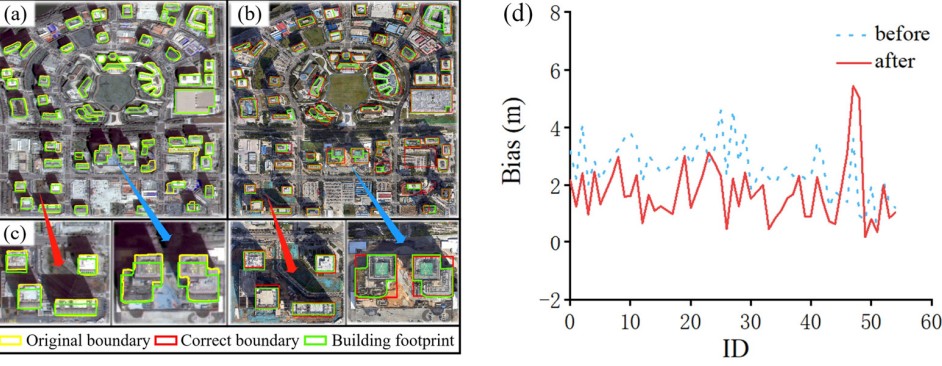

**Figure 13.** (**a**) Study area c before adjustment. (**b**) Study area c after adjustment. (**c**) Local detail diagram before and after adjustment. (**d**) Map of horizontal positioning accuracy.

As shown by the local detail effects illustrated in Figures 12c and 13c, the building outline obtained fits the reference outline well. This further demonstrates that the building outline adjustment algorithm proposed in this paper can improve the projection difference of high-rise buildings and obtain an accurate building footprint. The improvement in the precision of the building outline is closely related to the boundary precision of the initial outline and the measurement precision of the building height, which indirectly proves that the building height measurement precision in this study is relatively high. To further evaluate the precision of the building-outline-adjustment algorithm, the geometric center-of-mass deviation between the building outline and real building footprint before and after the algorithm processing is shown in Figures 12d and 13d. This method significantly improves the geometric center-of-mass deviations of buildings. Some results with large deviations are obtained, mainly because the building shadows are completely obscured. In this case, the height measurement error of the building increases significantly, resulting in a large deviation in the building outline correction process. Moreover, by adjusting only one side of the building boundary, it can be roughly estimated using Equations (4) and (6). The geometric center-of-mass deviation is improved by approximately 1 m, and the boundary precision is improved by approximately 5 m.

### 4.4. Analysis of Performance

The spatial resolution of the image also affects the precision of the extraction. The higher the resolution, the more precise the extraction of the shadow area of the building, and the higher the precision of the acquired shadow feature line length. In addition, the imaging time also affects the quality of the shadows. Therefore, this study further validates the performance of the method on different spatial resolution images and imaging times by selecting high-rise buildings for the experiments. Two sets of experiments are developed: the first is for the same area, same sensor, and different times, as shown in Figure 14a–e; the second is for the same area and different sensors, as shown in Figure 14f–i.

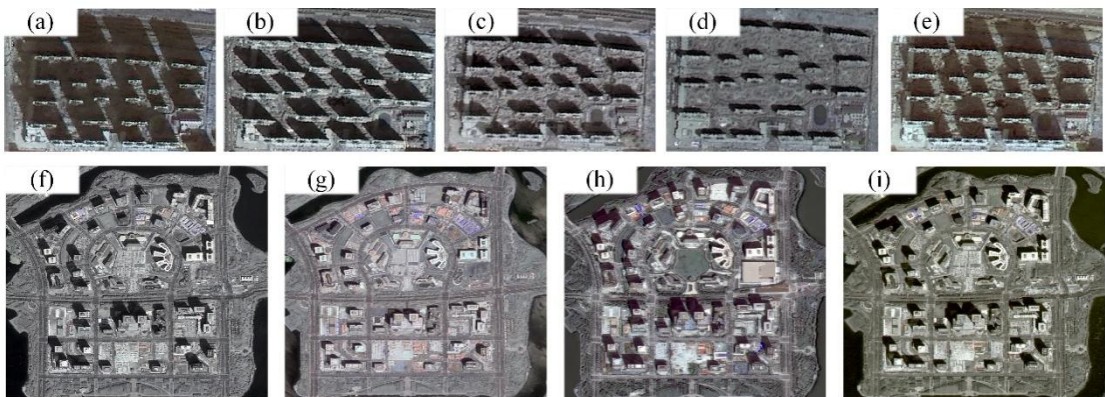

**Figure 14.** (**a**) 20210112. (**b**) 20230314. (**c**) 20210408. (**d**) 20200625. (**e**) 20211030. (**f**) GF1_20200911. (**g**) GF2_20200607. (**h**) GF7_20200920. (**i**) ZY3_20200904.

Figure 15 and Table 3 present the error statistics for the two sets of experiments. According to existing studies, it is a widely recognized fact that the length of shadows changes over time. Shadows are shortest during the summer months and longest during the winter months. The length of the shadows is positively correlated with the likelihood of being obscured; that is, the longer the shadow, the greater the likelihood. For the images of five different moments in the first set of experiments, it can be observed that in Figure 14a,e, the shadows are heavily obscured, and the error curves fluctuated with low precision, whereas the error curves of the remaining three moments are relatively stable. Theoretically, the higher the image resolution, the higher the precision in the second set of experiments. By combining the results in Figure 15b and Table 3, it can be observed that the error curve of GF2 fluctuated more and had the most prominent error. The error

curve of ZY3 exhibits less fluctuation and a relatively small error, which is inconsistent with expectations. After thoroughly analyzing this phenomenon, it is found that in addition to the image resolution, there exists another critical factor, the proportionality coefficient k, between the building height H and the shadow length L, and k has an inverse relationship with L, which is relative to Equation (1). According to error propagation theory, the height error of a building is the product of the shadow measurement error and the scale factor. When k increases, the height error of the building increases accordingly, which includes errors in the solar parameters and shadow measurements. The scale factors corresponding to Figure 14f–i are calculated as 1.501, 3.232, 1.362, and 1.599, respectively. GF2 has a larger scale factor and shorter shadows, resulting in a larger error in the building height.

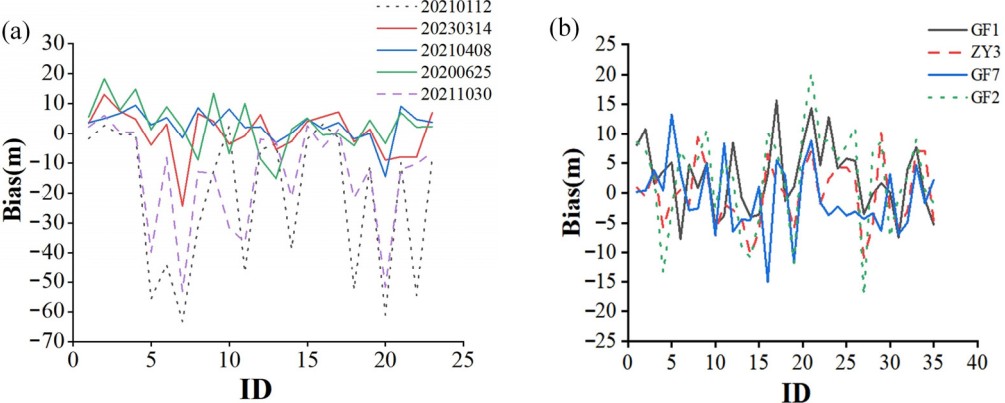

**Figure 15.** (**a**) Error statistical results in different periods. (**b**) Error statistical results in different sensors.

**Table 3.** Performance analysis.

| Test | Data | Time | MAE | RMSE |
|------|------|------|-----|------|
| 1 | GF1 | 12 January 2021 | 22.316 | 23.063 |
|   | GF1 | 14 March 2023 | 6.235 | 4.706 |
|   | GF1 | 8 April 2021 | 6.648 | 4.988 |
|   | GF1 | 25 June 2020 | 4.636 | 3.420 |
|   | GF1 | 30 October 2021 | 15.381 | 15.918 |
| 2 | GF1 | 11 September 2020 | 5.180 | 3.827 |
|   | GF2 | 7 June 2020 | 7.288 | 4.468 |
|   | GF7 | 20 September 2020 | 4.681 | 3.382 |
|   | ZY3 | 4 September 2020 | 4.940 | 3.766 |

In practical applications, the precision and operational efficiency of the algorithm must be addressed. To better evaluate the algorithm proposed in this paper, we compare the time with the traditional dense matching algorithm [35] in three study areas and obtain the time statistics, as shown in Table 4. It can be observed that the new algorithm has a shorter running time than the dense-matching algorithm, further proving its advantage in terms of running efficiency. Although the method proposed in this study is less accurate than the results of dense matching, it can quickly and accurately obtain 3D city data at a low cost to meet the needs of urban spatial planning, urban environmental assessment, and other applications.

**Table 4.** Time cost.

| Study Area | Number of Buildings | Time (min) | |
|------------|---------------------|------------|--|
|            |                     | Method in This Paper | Dense Matching |
| a | 1511 | 7.69 | 81.23 |
| b | 800 | 2.08 | 6.04 |
| c | 55 | 0.78 | 1.79 |

## 5. Discussion

This study utilizes shadow information to realize the building height measurement of a single remote sensing image, converting the shadow noise in the image into valuable information. Based on previous research, improving shadow extraction and shadow measurement and, at the same time, reducing the requirement for shadow integrity, the method proposed in this paper can maintain good versatility and practicability when the boundary is not smooth, the shadow is partially obscured, or the shadow is sticky. Although the method used in this study improves the precision and practicability of building height measurement using the shadow method, the influence of shadow formation conditions and other environmental factors on the shadow integrity of buildings remains a significant obstacle to measurement precision. Therefore, it is necessary to discuss the limitations of using shadow information to assess building height measurements from a single optical image.

(1)  Limitations of solar elevation angle and building height: The solar elevation angle ranges from $0°$ to $90°$ and occurs at $90°$ between the Tropic of Capricorn and the Tropic of Cancer. In most areas of China, however, the sun does not appear in the zenith direction. The solar elevation angle directly affects the length and direction of shadows. The longer the shadow length, the better the detection effect, and the richer and more complete the information about the building contained, whereas a shorter shadow length will easily lead to a lack of information, which is not conducive to the detection of the building. The impact on the building height measurements is explored by quantifying the change in k at noon throughout the year, as shown in Figure 16a. The k-value also represents the building height error due to the shadow length error of one pixel, which is statistically analyzed to yield a measurement error of approximately three pixels for the shadow. Under the condition of the same measurement error, the variation in the k value also reflects the uncertainty of the building height measurement, and a larger k value implies that the shadow length error has a larger effect on the building height estimation. From Figure 16a, it can be seen that building height measurements using images captured in summer have high uncertainty, and cloudy summer images are not of high quality.

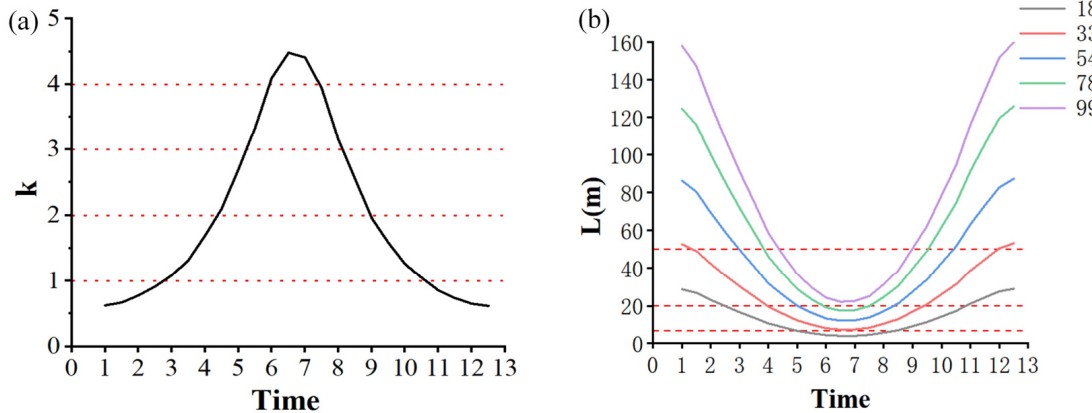

**Figure 16.** (**a**) Proportionality coefficient. (**b**) Shadow length.

However, the shadow method is not applicable to all building types. The shadow method is suitable for both high-rise and regular buildings. However, for super-high-rise buildings such as TV stations and skyscrapers as well as short-rise buildings such as bungalows, villas, and factories, the shadow method is less precise and may even fail. The numbers of floors in the typical residential designs are 6, 11, 18, 26, and 33. It is converted to a building height of 3 m per floor to investigate the changes in shadow length over a year. To further explore the range of applicability of the shadow method, the limit of

the complete projection of shadows is set to 50 m, and the initial condition for shadow detection is the length of shadows corresponding to 10 image pixels. Figure 16b shows the variation in shadow lengths for different heights over a year. It uses three red lines to indicate three constraints: the 50-m scale, the ZY3 initial condition scale, and the GF7 initial condition scale. According to the results, GF7 has a wider range of applications than ZY3 and is only valid for high-rise buildings. Considering the effect of the k value on building height estimation, with the likelihood of image shadows being obscured, it is believed that images captured around April and September are more suitable for building height measurement using the shadow method.

(2)　Because of restrictions on the building area, urban buildings tend to be denser to achieve optimal land utilization. This densification results in shorter spacing between neighboring buildings, leading to shadows sticking to each other and being unable to be projected fully onto the ground. The method in this study is based on building outline data for the zonal measurement of shadows, which reduces the requirement of shadow integrity but still does not solve the case of complete occlusion at the top of the shadow. In addition, the shadow areas of taller buildings may mask the lower ones, leading to difficulties in boundary extraction and shadow length measurement of the latter. The shadows of two buildings may be mixed, leading to an increase in the shadow measurement error of the taller building. The method used in this paper cannot recognize this situation, which is beyond the capability of semantic segmentation and shadow length measurement, and therefore cannot be solved.

(3)　Dramatic Terrain Undulations. Terrain undulations can affect the variation in shadow lengths, and the shadow method generally makes idealized assumptions regarding the terrain in building height calculations. However, in areas with high topographic relief (e.g., Chongqing), using images for shadow length measurements can lead to systematic deviations in the measured lengths from the actual lengths. This bias is a limitation of the shadow method in building height calculations and stems mainly from data limitations.

(4)　Complex Buildings. The shadow method has obvious advantages in measuring buildings with simple geometries; however, for buildings with complex structures and irregular areas, the accuracy of estimating the building height using the shadow method alone is low. Moreover, the shadow method requires that the top of the building be as flat as possible and aligned with the bottom. Height information and geometric boundaries are less applicable to the reconstruction of complex buildings. Therefore, it is necessary to combine other measurement techniques or use multiple data sources to improve the accuracy and reliability of the height estimation.

## 6. Conclusions

"Monolithic stereo" provides a new research idea for 3D modeling research. This study utilizes shadow information to complete the measurement of building height. By improving and refining the extraction and measurement algorithms in the shadow method, the precision and effectiveness of building height measurement in the shadow method are improved. Combined with deep learning building extraction technology, automated 3D reconstruction of urban high-rise buildings based on a single optical satellite image is realized. The experimental results show that the proposed method can reconstruct buildings in an image more quickly, automatically, and intelligently while maintaining geometric precision. The large-scale building entity visualization achieved using simplified and abstracted building models is more in line with people's impressions of the city and can also help users browse urban 3D environments and obtain urban spatial information accurately and quickly.

Given the limited information provided by single-view images, the estimation of building heights often suffers from error accumulation, leading to high uncertainty in the overall height estimation. With the continuous development of computer vision and deep learning, it can be combined with more proven techniques to improve the accuracy and

reliability of building height estimation, such as monocular depth estimation and nerve and building sideline measurements.

**Author Contributions:** Conceptualization, Zhixin Li, Song Ji, and Dazhao Fan; methodology, Zhixin Li; software, Song Ji; validation, Zhixin Li and Dazhao Fan; formal analysis, Song Ji; investigation, Fengyi Wang; resources, Zhen Yan; data curation, Ren Wang; writing—original draft preparation, Zhen Yan; writing—review and editing, Ren Wang; visualization, Fengyi Wang; supervision, Song Ji; project administration, Song Ji. All authors have read and agreed to the published version of the manuscript.

**Funding:** This research was supported by National Natural Science Foundation of China (No.42371459). Songshan Laboratory Project (No. 221100211000-5). National Science Foundation of Henan Province under Grant (No. 222300420592). High-resolution remote sensing, surveying, and mapping application demonstration system (Phase II) (No. 42-Y30B04-9001-19/21).

**Data Availability Statement:** Restrictions apply to the availability of these data.Data were obtained from the Natural Resources Satellite Remote Sensing Cloud Service Platform and are available at http://114.116.226.59/chinese/normal/ (accessed on 19 February 2024) with the permission of the Natural Resources Satellite Remote Sensing Cloud Service Platform.

**Acknowledgments:** We would like to thank Editage (www.editage.cn (accessed on 11 November 2023)) for English language editing.

**Conflicts of Interest:** The authors declare no conflict of interest.

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
