# Peer review of "Reconstruction of 3D Information of Buildings from Single-View Images Based on Shadow Information"

_ijgi, doi:10.3390/ijgi13030062_

Round 1
Reviewer 1 Report
Comments and Suggestions for Authors
The article is well done and presents a method for measuring building height from single-view images that integrates shadow information and semantic segmentation via deep learning on single-view remote sensing images. I appreciated the presentation and discussion of the results, which I found very clear. However, I have some comments that could improve the readability of the article:
1. In the title, the term 'three-dimensional reconstruction' is used. This expression refers more to the reconstruction of a 3D model, which seems limited in the article. I would suggest using 'Reconstruction of 3D information' or 'Reconstruction of building height information' in the title instead.
2. In the Introduction and state of the art section, some phrases related to the related work are not sufficiently supported by existing bibliography. I would recommend adding the appropriate references, for example, in Lines 57-60, 67-68 (where can we find the term 'shadow method'?), Line 95 (other methods?).
3. Lines 77-108. When the authors discuss their work, they should speak in the present tense, not the past, to clearly distinguish what they are currently doing from what has been done by others in the past. This also applies to the Methodology, Results, and Discussion sections.
4. Lines 182-183 and subsequent, Table 1 and elsewhere in the text. What do the authors mean by the acronyms GF and ZY? It's not clear.
5. Lines 189-190. How were the data collected? By which survey method?
6. Figure 1. The text of the Figure should be translated from Chinese to English.
7. Figure 3. In the sample data, at least from what is visible here, no taller building casts a shadow on a shorter building. In other words, no building shadows another shorter building. Since this situation could occur in reality, how was it addressed in this case? Was this case completely excluded, which could mean that in the future, the algorithm's application might create misclassified labels for this case, or was it somehow considered? Please discuss further.
8. Lines 285-288 and Figure 4a-4b. Figure 4a and 4b in the text do not correspond to the text in the caption.
9. Line 466 – which software was used?
Comments on the Quality of English Language
Moderate editing of the English language is required.
Reviewer 2 Report
Comments and Suggestions for Authors
The paper proposed a new method for 3D building reconstruction based on single-view remote sensing images. The proposed method combined shadow information to improve the segmented boundary of high-rise buildings. The experiments show the accuracy improvement of the proposed method. In general, the paper is well organized and clearly stated in a academic manner. The figures’ quality and the titles of each sub figures need be modified. Some figures with the remote sensing images as background are very vague.
Figure 7(e): what extraction results corresponding to (c)?
Figure 10 and the following figures: Basis and Bias?
The legends is a little ambiguous, e.g., correct boundary, building footprint,center line and so on.
Comments on the Quality of English LanguageMinor editing of English language required
Author Response
Please see the attachmen.

Reviewer 3 Report
Comments and Suggestions for Authors
The manuscript is about three-dimensional reconstruction of buildings from single-view images based on shadow information. The study proposes a method for the 3D reconstruction of single-gidiÅŸ remote sensing images, combining a shadow method with deep-learning semantic segmentation technology. Meanwhile, the proposed approach includes a novel algorithmic technique for building outline correction.
The literature review is covered in a way that critically engages with the state-of-the-art research methods ranging from shadow segmentation processes to depth-estimation techniques for the generation of digital surface models. It mainly distinguishes the threat of information overload during 3d reconstruction of cities from reliable and accurate building height estimation. However, the literature review does not emphasises sufficiently the research gap that the proposed method of building outline correction algorithm can fit in.
The paper exhausts a tailored methodological framework for accurate and reliable 3d reconstruction of realistic city models with limited information, testing the precision of the results with multiple datasets of single-view images. But the methodology part does not mention what specialties make the study area (Zhengzhou, China) got selected and why not other cities. The most important part of the methodology is the section “3.5.2. Correction of Building Outline” which requires better explanation as some of the formulas are not in order with the text flow. Thereby, the figure 6 may be placed to the beginning of the “3.5.2” section. Eq. 4 and 5 need to be cited and explained within text prior to equations.
The strongest part of the paper is the analysis section divided into qualitative and quantitative subsections. In this section, the placement of the figures (7 to 14) and the table (2) should be corrected so they are visible just after being cited within the text.
The weakest part of the paper is that the results of calculations that lead to mean absolute error and root mean square error are not given, such as deviations of centre of mass. They are directly interpreted into percentage outcome on Table 2, which is good but the readers would likely seek and easily find the information regarding the calculation results of the proposed algorithms.
The discussion nicely starts with an entry sentence claiming a real discussion around converting shadow noise in the image into valuable information. But instead of amplifying the research findings into implications at the more though-provoking level, it continues with the limitations of the research that ends with the repetition of information such as the challenges of undulating terrains, complex buildings and building forms that does not provide applicable shadow data.
The conclusion need to be improved. The beginning sentence starts with the quotation of ”monolithic stereo” which is irrelevant to the flow of the text. The conclusion states that “the proposed method can reconstruct buildings in an image more quickly … while maintaining geometric precision.” It is understood that the paper benefits from the interchangeable use of words; accuracy and precision within the text. The paper also tests precision results, but only to show the level of accuracy. The proposed method does not maintain the precision but the accuracy. It is proven to be one of the methods to reach satisfactory level of accuracy by testing how precise it can be, but not maintaining %100 precision.
More specific comments:
Page 1, Line 13: “… costly for low-cost …” (better use of words or rewriting required here).
Page 1, Line 31: Introduction suddenly mentions something related to the context of the study. If it is necessary to mention China’a case, give more information regarding how other parts of the world are doing.
Page 2 and 3: The list of contributions needs elaboration and possibly rewriting. The claimed contribution 2 is part of the first one. They can be combined. The third one sounds a methodological phase rather than a contribution. There is an unnecessary stress that this part of the paper produces regarding the number of contributions. It be relieved by concisely elaborating the first and only strong contribution of this hardwork.
Page 2, Line 94: Do you mean “… the challenge of 3D reconstructing high rise buildings in cities.”
Page 2, Line 95, the last sentence of the paragraph: It is not easy to understand when you mention “a simple principle.” If you remove it, it becomes clearer.
Page 3, Line 104-108: This part requires certain elaboration on what the sections are all about. There is lack of information.
Page 4, Line 189: Do you mean “Realistic 3D models”?
Page 5, Line 197: The paper may benefit from reconsidering the use of the term “morphological structure” as morphology studies may not totally agree with this definition.
Page 6, Line 243: The last sentence of the paragraph has a claim that requires citation.
Page 6, Line 246 and 247: There are two consecutive sentences that repeats the same verb “show.” It will benefit from differentiation.
Page 8, Line 277: The first sentence of the page states that the proposed algorithm is developed from a previous study which should be mentioned in the abstract.
Page 8, Line 283: Delete “As shown in Figure 4,”
Page 8, Line 292: Add view after schematic. “Schematic view of the geometric…”
Page 9, Line 320: “… elevation angle, H is the building elevation angle…” H is not shown in Figure 5. And do you mean elevation height, instead of elevation angle?
Page 10: Eq. 5 and 6 need to be explained within the text.
Page 11, Line 367: “ The paragraph does not starts with a proper sentence.
Page 11, Line 381: Here states the use of ArcGIS and QGIS but not mentions why two similar software solutions are utilised or how their use are distinguished.
Page 12, Line 401 to 403: This sentence is better to proceed Eq. 12.
Page 16: Figure 11 may benefit from giving references to Liasis and Stavrou (Year) and Xie (Year).
Page 16, Line 512: The abbreviation of digital orthophoto map (DOM) is usEd twice throughout the manuscript. So it is better to spell it out every time.
Page 20, Line 607: “In most areas of China” suddenly appears. Try to interpret the results from a global perspective and then emphasise the local facts.
Page 23, Line 766: The reference [35] has not provided page numbers.
Comments on the Quality of English LanguageMinor editing is required, especially with regards to the choice of words in the right place. See comments and suggestions.
